# Alteration of Colonic Bacterial and Fungal Composition and Their Inter- and Intra-Kingdom Interaction in Patients with Adenomas with Low-Grade Dysplasia

**DOI:** 10.3390/microorganisms11051327

**Published:** 2023-05-18

**Authors:** Ding Heng, Min Zhang, Yuhan Yuan, Xinyun Qiu

**Affiliations:** 1Department of Gastroenterology, The First Affiliated Hospital of Nanjing Medical University, No. 300, Guangzhou Road, Nanjing 210029, China; yznyhd@yeah.net (D.H.); zhangmin0717@126.com (M.Z.); 2Department of Endoscopic Center, The First Affiliated Hospital of Nanjing Medical University, No. 300, Guangzhou Road, Nanjing 210029, China; yuanyuhan512@126.com; 3F. Widjaja Inflammatory Bowel and Immunobiology Research Institute, Cedars-Sinai Medical Center, Los Angeles, CA 90048, USA

**Keywords:** intestinal bacteria, intestinal fungi, colorectal adenomas, low-grade dysplasia

## Abstract

Colorectal cancer (CRC) develops from pre-cancerous cellular lesions in the gut epithelium and mainly originates from specific types of colonic adenomas with dysplasia. However, gut microbiota signatures among sampling sites in patients with colorectal adenomas with low-grade dysplasia (ALGD) and normal control (NC) remain uncharacterized. To characterize gut microbial and fungal profiles in ALGD and normal colorectal mucosa tissues. We used *16S* and *ITS1-2 rRNA* gene sequencing and bioinformatics analysis on the microbiota of ALGD and normal colorectal mucosa from 40 subjects. Bacterial sequences in the ALGD group showed an increase in *Rhodobacterales, Thermales, Thermaceae, Rhodobacteraceae*, and several genera, including *Thermus, Paracoccus, Sphingobium*, and *Pseudomonas,* compared to the NC group. Fungal sequences in the ALGD group showed an increase in *Helotiales, Leotiomycetes*, and *Basidiomycota*, while several orders, families, and genera, including *Verrucariales, Russulales*, and *Trichosporonales,* were decreased. The study found various interactions between intestinal bacteria and fungi. The bacterial functional analysis showed increased glycogen and vanillin degradation pathways in the ALGD group. Meanwhile, the fungal functional analysis showed a decrease in pathways related to the biosynthesis of gondoate and stearate, as well as degradation of glucose, starch, glycogen, sucrose, L-tryptophan, and pantothenate, and an increase in the octane oxidation pathway in the ALGD group. The mucosal microbiota in ALGD exhibits altered fungal and microbial composition compared to the NC mucosa, potentially contributing to the development of intestinal cancer by regulating specific metabolic pathways. Therefore, these changes in microbiota and metabolic pathways may be potential markers for diagnosing and treating colorectal adenoma and carcinoma.

## 1. Background

Colorectal cancer (CRC) is the third most common cause of cancer-related deaths [1]. Approximately 5% of the population suffers from CRC, and 90% of cases occur after the age of 50 years [2]. Most CRCs originate from specific types of colonic adenomas, which result from sporadic mutations in the colorectal polyp pathway or DNA mismatch repair and, by definition, include low-grade dysplasia [2]. Since the 1960s, the microbiome has been hypothesized to play a role in CRC carcinogenesis. Bacterial strains are altered during adenoma-carcinoma development [3,4], and fungal strains, including Malassezia and Candida, are aggravated in the gut of patients with CRC [5]. Most studies have compared the intestinal microbiome of healthy individuals to those with CRC or colonic adenomas using stool samples. However, the luminal microbiome is usually not a real colonizer but could instead be a transient environmental microbiota which cannot adhere to the colonic mucosa and interact with the mucosal immune system [6,7]. Therefore, studying the microbiota composition of mucosal samples is crucial. However, it’s important to note that microbiota profiles can vary significantly between individuals due to various factors, such as diet, environment, immune responses, medication usage, etc. Therefore, the findings on gut microbiota are inconsistent across previous studies.

The gut microbiota comprises bacteria, fungi, viruses, and protozoa [8]. Bacteria have received the most attention due to their abundance and ease of isolation, culture, and identification. However, technological advancements have revealed that fungi and other microorganisms also play crucial roles in gut health and disease [9]. In the past, fungi were often overlooked in research due to their difficulty in identification and characterization, and previous studies focused mainly on bacterial communities [9]. Consequently, their significance in gut health and disease may have been underestimated or ignored.

To better understand the role of microbiota in the progression toward CRC, it is important to explore the microbiota in direct proximity to the epithelium. Therefore, obtaining mucosal specimens from precursor lesions and tumors could provide greater insight into mucosa-associated microbiota. In this study, we conducted a metagenomic profiling analysis of both bacterial and fungal microbiota in colonic mucosal samples from patients with adenomas with low-grade dysplasia (ALGD) and normal controls (NC). We aimed to characterize the different microbiota composition of adenoma mucosa against their normal control (NC) and explore inter- and intra-kingdom correlations between intestinal bacteria and fungi. This can provide insight into colorectal tumorigenesis.

## 2. Methods

### 2.1. Patients and Biopsy Sample Collection

Colorectal biopsies were collected at the Endoscopic Center of the First Affiliated Hospital of Nanjing Medical University. All subjects underwent colonoscopy for screening and were found to have colonic polyps. Specimens from polyps and the adjacent normal control mucosa were biopsied and stored at −80 °C for further analysis. All of the included participants finished the complete colon examination, and subjects with the following conditions were excluded: (1) with intestinal inflammation or had a history of pigmentation polyposis syndrome, familial intestinal polyps; (2) with gastroenterology surgery history; (3) with malignant tumors and cachexia, or a history of radiation or chemotherapy; (4) Female subjects who were pregnant or lactating. (5) less than 18 years old; (6) participants had taken antibiotics, anti-fungal agents, probiotics, or prebiotics for at least eight weeks before enrollment. Written informed consent was obtained from all participants. In addition, the clinical characteristics of the participants were recorded, including age, sex, disease history, pathological pattern, and histology. The study protocol was approved by the Medical Ethics Committee of the First Affiliated Hospital of Nanjing Medical University (Approval Code: 2020-5R-412).

### 2.2. DNA Extraction, PCR Amplification, and 16S rRNA Gene Sequencing

The E.Z.N.A.^®^ Stool DNA Kit (D4015, Omega Inc., Norwalk, CT, USA) was utilized to extract total genomic DNA from mucosal biopsy samples, following the manufacturer’s protocol. The eluted total DNA was stored at −80 °C before PCR analysis, with an elution buffer volume of 50 μL. Validation of DNA concentration and quality was accomplished via a NanoDrop spectrophotometer (Thermo Fisher Scientific, Waltham, MA, USA) and 2% agarose gel (Tanon, Shanghai, China). Nuclease-free water control was used as a blank control.

The universal primers 341F (5′-CCTACGGGNGGCWGCAG-3′) and 805R (5′-GACTACHVGGGTATCTAATCC-3′) were utilized to perform PCR analysis of the V3-V4 variable regions of the bacterial *16S rRNA* gene. For amplification of the eukaryotic (fungi) small-subunit rRNA gene’s ITS2 region, the ITS1FI2 (5′-GTGARTCATCGAATCTTTG-3′) and ITS2 (5′-TCCTCCGCTTATTGATATGC-3′) primer sets were employed. Specific barcodes for each sample and sequencing universal primers were attached to the 5’ ends of the primers.

A 25 μL reaction mixture was utilized for the PCR reaction, incorporating 25 ng of template DNA, 12.5 μL of High-Fidelity PCR Master Mix (NEB), 2.5 μL of each primer (1 μM), and PCR-grade water as needed to reach the proper volume. To amplify the prokaryotic 16S fragments, the PCR conditions consisted of an initial denaturation step at 98 °C for 30 s, followed by 32 cycles for *16S rRNA* (36 cycles for *ITS1-2 rRNA*). These cycles included a denaturation stage at 98 °C for 10 s, an annealing stage at 54 °C for 30 s, and an extension stage at 72 °C for 45 s. The final extension was conducted at 72 °C for 10 min. The PCR products were verified through 2% agarose gel electrophoresis (run at a constant voltage of 120 V for 40 min). Regarding DNA extraction, ultrapure water was employed as a negative control to exclude the possibility of positive PCR results.

The amplicon was purified with AMPure XP beads (Beckman Coulter, Carlsbad, CA, USA) and subjected to a secondary PCR reaction. The final amplicon was quantified using a Qubit dsDNA assay kit (Life Technologies, Carlsbad, CA, USA). A sequencing library was generated using the TruSeq^®^ DNA PCR-Free Sample Preparation Kit (Illumina, San Diego, CA, USA) as per the manufacturer’s instructions and assessed on the Qubit@ 2.0 Fluorometer (Thermo Fisher Scientific, Waltham, MA, USA) and Agilent Bioanalyzer 2100 system (Agilent Technologies, Santa Clara, CA, USA). The library was sequenced on an Illumina NovaSeq (PE250) high-throughput platform, and 250 bp paired-end reads were generated. The PCR products were purified by the Qiaquick gel extraction kit (Qiagen Hilden, Germany). Amplicon pools were prepared for sequencing, and the size and abundance of the amplicon library were evaluated with an Agilent 2100 Bioanalyzer (Agilent, Santa Clara, CA, USA) and Library Quantification Kit for Illumina (Kapa Biosciences, Woburn, MA, USA), respectively. The sequencing of libraries was performed on the NovaSeq PE250 platform. After removing the background, the feature list and sequence were obtained, and a non-metric multidimensional scaling (NMDS) analysis was conducted.

### 2.3. Bioinformatic Analysis

The samples were subjected to sequencing on an Illumina NovaSeq platform using a paired-end 250 run. Unique barcodes were used to assign paired-end reads to different samples, then merged with FLASH. First, quality filtering was performed on the raw reads, and high-quality clean tags were obtained with fqtrim (v0.94). Next, the chimeric sequences were removed using Vsearch software (v2.3.4), and subsequently, the feature table and feature sequence were generated using the DADA2 pipeline.

Alpha and beta diversity were determined by randomly normalizing the same sequences and analyzing them with QIIME2. The resulting data was used to generate graphs using an R package. Alpha diversity was applied using richness (Chao1 index) and diversity (Shannon and Simpson index) indices. Beta diversity analysis was employed to compare groups among samples, and non-metric multidimensional scaling (NMDS) analysis and principal coordinate analysis (PCoA) was carried out using a weighted UniFrac distance matrix. Linear discriminant analysis (LDA) effect size [10] was used to identify differentially abundant bacterial taxa associated with groups of participants. The LDA value threshold was set at two, and the *t*-test and Wilcoxon signed-rank test were employed to assess the significance using QIIME (version 1.9.0). Prediction of metagenomic functions was carried out using Phylogenetic Investigation of Communities by Reconstruction of Unobserved States (PICRUSt2) based on the normalized bacterial and fungal OTU tables [11]. Pathways differing in abundance between the ALGD and NC groups were obtained using Welch’s *t*-test, and the Storey FDR was used to correct for multiple tests. 

### 2.4. Correlation Analysis

Pearson’s correlation was used to test whether a relationship existed between bacterial and fungal diversity on Shannon’s index and taxon relative abundance. The correlation between bacterial and fungal taxon relative abundance in the ALGD group was measured by distance correlation and as described previously [12].

### 2.5. Statistics

Differences in the Shannon, Simpson, and Chao1 diversity indices between the two groups were analyzed using the Wilcoxon test. Intergroup differences at the phylum, class, order, family, genus, and species level in each cluster were analyzed by the linear discriminant analysis (LDA) effect size (LEfSe) method [13] with default settings on the Galaxy/Hutlab website (https://huttenhower.sph.harvard.edu/galaxy/root, accessed on 1 January 2023). The correlations between Shannon, Simpson, and Chao1 diversity indices were measured using Spearman’s rank coefficients. Similarly, the correlation between bacterial and fungal taxa at the phylum and genus level was also measured using Spearman’s rank correlation coefficient. Mann-Whitney U test was used to determine statistically significant differences between bacterial and fungal relative abundance in ALGD and NC samples. A *p*-value < 0.05 was considered statistically significant.

## 3. Results

### 3.1. Charistics of Subjects

A total of 117 participants were initially enrolled in this study. However, 44 participants were first excluded due to malignant tumors and cachexia history (*n* = 12), gastroenterology surgery history (*n* = 5), pregnant or lactating (*n* = 2), or taken antibiotics or probiotics (or prebiotics) within eight months (*n* = 25) before the colonoscopy. Subsequently, an additional 33 participants were excluded due to the presence of intestinal inflammation (*n* = 6) or their intestinal specimens being diagnosed as hyperplastic polyps (*n* = 15), high-grade intraepithelial (*n* = 8), or neoplastic tumors (*n* = 4) during colonoscopy. Forty (aged between 22–56 years, 18 females and 22 males) subjects diagnosed with colonic adenoma with low-grade dysplasia were included (Appendix A).

### 3.2. Global Shifts in the Colonic Microbes

To observe the changes in the colonic microbes in colorectal ALGD and their adjacent normal control tissue, we analyzed sequencing on 80 colonical mucosal biopsies (40 ALGD and their NC) from 40 patients. An average of 98,250 high-quality paired reads were obtained for each sample, and operational taxonomic units (OTUs) were defined using a 97% similarity cut-off value. The Venn diagram showed that 2947 OTUs were generated from ALGD samples and 3131 OTUs were generated from the NC, with 1520 OTUs shared between both groups, indicating a greater bacterial diversity in the NC colonic mucosa (Figure 1A). The rarefaction curve exhibits that the current sequencing depths were sufficient, and no new OTUs were detected when the sequencing depth was increased (Figure 1B). The bacterial composition did not differ significantly between ALGD and NC groups using the PCoA (Figure 1C). To further evaluate the alpha diversity and beta diversity in both groups, the Chao1, Simpson, and Shannon index between groups did not show remarkable differences between groups, and the beta diversity analysis did not show distinct clusters between groups (Figure 1D–F). Interestingly, the Chao1, Simpson, and Shannon index in the ALGD group was positively correlated with that in the NC (Figure 1G–I).

We further analyzed the colonic bacterial composition of the two groups. The bacterial community from the 40 participants’ mucosal samples was classified into 33 phyla, 104 classes, 217 orders, 352 families, and 815 genera (Appendix A). At the phylum level, the intestinal microbiome was dominated by *Proteobacteria* (47.71%), *Firmicutes* (22.34%), *Bacteroidota* (17.14%), and_*Fusobacteriota* (8.50%) (Figure 2A). When analyzed at the genera level, sixteen genera, namely *Escherichia-Shigella*, *Bacteroides*, *Fusobacterium*, *Faecalibacterium*, *Haemophilus*, *Citrobacter*, *Klebsiella*, *Intestinibacter*, *Streptococcus*, *Actinobacillus*, *Sutterella*, *Prevotella*, *Roseburia*, *Prevotella_9*, *Ruminococcus torques* group, and *Acinetobacter* were identified above 1% on average (Figure 2B). Interestingly, no significant differences were observed in the major bacterial phyla and genera between the two groups (Figure 2B).

LEfSe analysis was used to determine the specific bacteria taxa associated with ALGD. Compared with the NC, the relative abundance of Acidaminococcus was significantly decreased in the Abiotrophia genus and species in the ALGD group (Figure 2C, D). However, the orders Rhodobacterales and Thermales; the families Thermaceae and Rhodobacteraceae; the genera *Thermus*, *Paracoccus*, and *Sphingobium*; and *Pseudomonas* species were increased in the ALGD group compared with that in the NC (Figure 2C,D).

### 3.3. Global Shifts in the Colonic Fungi

We further observed changes in colonic fungi in colorectal ALGD and adjacent NC tissues from the 40 participants. An average of 92,112 high-quality paired reads were obtained for each sample, and operational taxonomic units (OTUs) were defined using a 97% similarity cut-off value. The Venn diagram showed that 627 OTUs were generated from ALGD samples and 1124 OTUs were generated from the NC, with 294 OTUs shared between both groups, indicating a greater fungal diversity in the NC colonic mucosa (Figure 3A). The rarefaction curve showed that the current sequencing depths were sufficient, and no new operational taxonomic units (OTUs) were detected when the sequencing depth was increased (Figure 3B). The fungal composition did not differ significantly between ALGD and NC groups using the PCoA (Figure 1C). Concerning alpha and beta diversities in both groups, the Chao1 index was significantly decreased in the ALGD group compared with the NC group (Figure 3F), reveling the significant decrease of fungal species diversity in the ALGD group. In addition, the Simpson and Shannon indices between groups did not show remarkable differences and were not significantly correlated with each other between the groups (Figure 3D,E). Interestingly, the Chao1, Simpson, and Shannon index in the ALGD group did not significantly correlate with that in the NC (Figure 3G–I).

We further analyzed the composition of fungal bacteria in ALGD and NC groups. The fungal community from the 40 participants’ mucosal samples was classified into ten phyla, 37 classes, 84 orders, 183 families, and 327 genera (Appendix A). At the phylum level, the intestinal fungi were dominated by Ascomycota (72.91%), Basidiomycota (17.61%), and Zygomycota (4.44%) (Figure 4A). When analyzed on the genera level, twenty-one genera, namely *Alternaria, Malassezia, Microascus, Ascomycota_unclassified, Davidiella, Gibberella, Phialemonium, Actinomucor, Candida, Valsaceae_unclassified, Mortierella, Cladosporium, Phialocephala, Aspergillus, Cryptococcus, Talaromyces, Acremonium, Saccharomyces, Thermoascus, Lycoperdon*, and *Cyllamyces* were identified above 1% on average (Figure 4B). However, no significant differences were found in the major fungal phyla and genera (Figure 4B).

LEfSe analysis was used to identify the specific fungal taxa associated with ALGD. Compared with NC, the relative abundance of *Helotiales, Leotiomycetes*, and *Basidiomycota* increased at the order, class, and phylum levels, respectively (Figure 4C,D). While the *Verrucariales, Russulales*, and *Trichosporonales* were decreased at the order level, the *Didymellaceae, Pleosporales_Incertae_sedis, Eurotiales_Incertae_sedis, Verrucariaceae, Sordariales_unclassified, Entolomataceae*, and *Trichosporonaceae* were decreased at the family level; *Strelitziana*, *Thermomyces*, *Verrucariaceae*_unclassified, *Nectria*, *Neonectria*, *Sordariales*_unclassified, *Entoloma*, and *Trichosporon* were decreased at the genus level (Figure 4C,D).

### 3.4. Correlation between Intestinal Bacterial and Fungal Composition

We evaluated the relationship between intestinal bacterial and fungal taxa in the ALGD and NC groups at the phylum and genus levels. At the phylum level, Firmicutes showed a significant positive correlation with Bacteroidetes and a negative correlation with Proteobacteria (Figure 5A). Basidiomycota was negatively correlated with Ascomycota (Figure 5A). At the genus level, we observed a relationship between the 16 main bacteria (found in more than 50% of the samples and had a relative abundance >0.01) and 21 main fungal genera. The complex relationships between bacteria and fungi are illustrated in Figure 5B.

### 3.5. Functional Analysis

Bacterial functional analysis was conducted on colonic biopsy samples from the ALGD and NC groups. Four hundred thirty-six pathways were identified in both groups, four of which were significantly increased in the ALGD group (*p* < 0.05), including glycogen degradation II (eukaryotic), vanillin and vanillate degradation II, the super pathway of vanillin and vanillate degradation, and vanillin and vanillate degradation I. Therefore, these results suggest that alteration of bacterial function could be involved in the pathogenesis of ALGD (Figure 6A). In addition, the fungal functional analysis was also conducted on samples from both groups. Eighty-eight pathways were identified in both groups, eight of which were found to be significantly decreased in the ALGD group (*p* < 0.05), including gondoate biosynthesis (anaerobic), stearate biosynthesis II (bacteria and plants), glucose and glucose-1-phosphate degradation, starch degradation V, glycogen degradation I (bacterial), sucrose degradation III (sucrose invertase), L-tryptophan degradation to 2-amino-3-carboxymuconate semialdehyde, and pantothenate and coenzyme A biosynthesis (Figure 6B). However, the octane oxidation pathway was increased in the ALGD group compared with the NC (Figure 6B). 

## 4. Discussion

Most CRCs arise from specific types of colonic polyps (adenomas) which form due to spontaneous mutations or DNA mismatch repair in the adenomatous polyposis coli pathway, which contains low-grade dysplasia [2,14]. Low-grade dysplasia is an early precancerous change and, if left untreated, can develop into high-grade dysplasia or cancer. Therefore, proctocolectomy is recommended when it is identified. In clinical settings, adenomatous polyps are predictive tools for patients more prone to developing CRC. 

Thus, in this study, we identified colorectal adenoma-associated microbiota in ALGD patients and compared the microbiota in ALGD-associated mucosa to that of NCs. In previous studies, the microbiota profile differences were less significant than those reported between the CRC and NC groups [15]. However, there were still some differences between the ALGD and NC groups. First, we evaluated bacterial colonization in ALGD-associated mucosa and found that the OTU number was slightly lower than that in the NC. Species richness and community diversity were not markedly different between the ALGD and NC groups. This corroborated a study by Peter et al. [16], which suggests that the microbial species richness was significantly different only in advanced conventional adenoma cases.

Interestingly, the Chao1, Simpson, and Shannon indices showed a positive association between ALGD and NC tissues, exhibiting a close relationship of bacterial colonization between ALGD and adjacent normal colorectal tissue. We further studied the bacterial composition in the colon at the phylum level. Proteobacteria, Firmicutes, Bacteroidetes, and Fusobacteria were the four main phyla in the colonic mucosa. Proteobacteria was the most dominant phylum, outnumbering the other three.

Next, we studied the different bacterial compositions in the colon at the different taxa levels. Ten taxa, including uncultured *Pseudomonas* spp., *Rhodobacterales*, *Sphingobium*, *Rhodobacteraceae*, *Paracoccus*, *Thermus*, *Thermaceae*, and *Thermus* unclassified species, and *Thermales* order species were increased in ALGD group. In comparison, two taxa (an uncultured_*Abiotrophia*_species and *Abiotrophia* genus) were decreased in the ALGD group compared with NC. Among the twelve taxa, several taxa colonize the human gut. Wheatley et al. [17]. reported that a *Pseudomonas* species (*P. aeruginosa*) can translocate from the gut to the lungs and cause tissue damage during lung infection with greater pro-inflammatory gene expression. Mei et al. [18]. reported that *Sphingobium* was more abundant in the duodenal mucosa of patients with pancreatic cancer than in the duodenal microbiotas of healthy controls. As for *Abiotrophia*, one study reported its lower abundance in colonic mucosal tissue in high-fat-diet subjects [19], and another study reported that it is more abundant in the tongue coating microbiome in patients with gastric cancer [20]. Previous studies have demonstrated the pro-inflammatory function of these bacterial taxa in human disorders; however, the relationship between these bacterial taxa and ALGD development has not been reported.

We further investigated fungal colonization in mucosa-associated with ALGD and found that the number of observed fungal OTUs was slightly lower than in the NC group. The principal coordinate analysis (PCoA) results showed that the fungal communities in ALGD-mucosa were more separated than the NC-mucosa. We observed a decrease in species richness in the ALGD group, which is consistent with the decreased Chao1 index compared to the NC group. However, the two groups had no significant differences in the fungal community diversity (Simpson and Shannon indices). Furthermore, unlike the bacterial analysis, we found no significant association between ALGD and NC regarding Chao1, Simpson, and Shannon indices.

In this study, we studied the fungal composition in the colon at the phylum level. Ascomycota, Basidiomycota, and Zygomycota were the three predominant phyla in the colonic mucosa, with Ascomycota being the most dominant phylum, outnumbering the other two. However, it is essential to note that about 5% of fungi could not be identified at the phylum level, indicating a significant gap in our knowledge about fungi. Therefore, we further examined the fungal composition at different taxonomic levels. Twenty-two taxa, including *Didymella, Didymellaceae, Pleosporales Incertae sedis, Strelitziana, Thermomyces*, *Eurotiales Incertae sedis, Verrucariaceae* unclassified, *Verrucariaceae, Verrucariales, Nectria, Neonectria, Sordariales* unclassified, *Sordariales* unclassified, *Entoloma, Entolomataceae, Russulales, Trichosporon, Trichosporonaceae, and Trichosporonales* were decreased in the NC group compared with that in the ALGD group. However, the number of *Helotiales*, *Leotiomycetes*, and *Basidiomycota* increased in the ALGD group.

Although previous data did not show a direct relationship between most of the fungi mentioned above and the development of ALGD, some fungi have been associated with inflammation and metabolic pathways closely linked to colorectal cancer (CRC). For example, the relative abundance of the fungal genus *Thermomyces* has been found to increase in bronchoalveolar lavage fluid samples from patients with chronic obstructive pulmonary disease (COPD), which is also associated with an increased risk of CRC [21]. Feng et al. [22]. reported that *Trichosporon asahii* Y2 could increase lipase activity in the digestive tract and promote obesity and hyperlipidemia. Andrade et al. [23]. reported that *T. asahii* and *T. mucoides* could produce glucuronoxylomannan, which participates in several immunoregulatory mechanisms and causes inflammation. *Leotiomycetes* and *Helotiales* were decreased in newly diagnosed treatment-naïve children with Crohn’s disease compared with healthy controls, although the difference was not statistically significant [24]. *Thermomyces* are strongly associated with metabolic disturbances and weight gain [25].

The interactions between bacteria and fungi have been overlooked in previous studies. In this study, we observed relationships between the inter- and intra-kingdoms of bacteria and fungi. Our study has observed relationships between these two kingdoms at inter- and intra-kingdom levels. We found that, at the phylum level, there was a negative association between *Proteobacteria* and both *Firmicutes* and *Bacteroidetes*, as well as a negative association between *Ascomycota* and *Basidiomycota* in the LAGD group. These findings are consistent with previous studies that have reported an increase in the *Ascomycota/Basidiomycota* ratio in colon polyps and CRC patients compared to healthy controls [26]. We examined associations between bacterial and fungal taxa at the genus level, focusing on 16 main bacterial genera and 26 fungal genera. Previous research has linked several taxa to intestinal inflammation and/or tumor formation. Chen et al. [27]. The abundance of *Clostridium, Roseburia*, and *Eubacterium* spp.—genera involved in butyrate fermentation—was lower in the colorectal adenoma group compared to the healthy control group.

In contrast, Enterococcus, Streptococcus spp., and Proteobacteria phyla were enriched in the adenoma group. Keisuke et al. [28]. reported that patients with colorectal adenoma had higher levels of *Fusobacterium, Parvimonas*, and *Atopobium* than healthy individuals. *Fusobacterium spp*. was the most differentially abundant bacterial taxon in patients with colorectal adenoma, while the microbiome of healthy individuals had a higher abundance of *Lachnobacterium, Salmonella*, and *Moraxellaceae*. In another study by Chattopadhyay et al. [29], patients with CRC had higher levels of *Escherichia coli*, *E. faecalis*, *F. nucleatum*, and *Streptococcus gallolyticus*. At the same time, *Bifidobacterium, Clostridium, Faecalibacterium*, and *Roseburia* were less abundant. *Malasseziomycetes* (fungal class) was enriched in CRC patients, while *Saccharomycetes* (including *Lypomyces starkeyi* and *Saccharomyces cerevisiae*) and *Pneumocystidomycetes* were lower [26]. 

In this study, we found that *Fusobacterium*, which may play a role in the etiology of CRC due to its coaggregation ability, was significantly negatively associated with *Prevotella* 9 and *Acinetobacter* and was positively associated with *Themoascus*. *Faecalibacterium*, a well-known butyrate-producing bacteria that can protect the intestinal epithelial cells [30,31], was significantly positively associated with *Bacteroides*, *Sutterella*, *Roseuria*, and *Ruminococcus torques* 9, and negatively associated with *Citrobacter*. *Roseburia*, which can improve intestinal biodiversity, glucose tolerance, and rejuvenate colon cells [32], was positively associated with *Bacteroides*, *Faecalibacterium*, *Sutterella*, and *Ruminococcus torques* and negatively associated with *Citrobacter*. *Saccharomycetes*, which were reported to be decreased in patients with CRC, were positively associated with *Prevotella*, *Candida*, and *Lycoperdon* and negatively associated with *Escherichia-Shigella* and *Cladosporium*. *Candida*, enriched in gastrointestinal tumor samples and associated with inflammation and tumorigenesis [33], was positively associated with *Haemophilus*, *Actinobacillus*, *Prevotella*, *Phialocephala*, *Saccharomyces*, and *Lycoperdon,* and negatively associated with *Escherichia-Shigella*.

The functional prediction analysis provided insights into how the intestinal microbiota affects the initiation and development of ALGD. Using PICRUSt2, it was found that the vanillin and vanillate degradation pathways and the glycogen degradation pathways were increased in the ALGD group compared to the NC group. Vanillin, vanillic acid, and vanillyl alcohol have anti-inflammatory effects and can act on oxidative stress [34], which may contribute to the development of ALGD. In addition, glycogen degradation is a harmful pathway that can cause CRC, which was also increased in the ALGD group compared with the NC group [35], illustrating intestinal bacteria involved in the function of this pathway. 

When the intestinal fungi were analyzed using PICRUSt2, it was found that the glucose and glucose-1-phosphate degradation, glycogen degradation I (bacterial), gondoate biosynthesis (anaerobic), L-tryptophan degradation to 2-amino-3-carboxymuconate semialdehyde, pantothenate and coenzyme A biosynthesis I, starch degradation V, stearate biosynthesis II (bacteria and plants), and sucrose degradation III (sucrose invertase) pathways were decreased in the ALGD group compared to the NC group. On the other hand, octane oxidation was increased in the ALGD group. While the effects of these pathways in the initiation and development of ALGD/CRC have not been reported, studies have shown that the 2-amino-3-carboxymuconate semi-aldehyde degraded from L-tryptophan is a carcinogen that can cause tumor initiation [36]. Additionally, stearate can act as an energy source for the growth of specific cancer cells. Stearate biosynthesis II is reported to be involved in the development of certain types of cancer [37]. Although there is currently no direct link between sucrose degradation and cancer, research suggests that diets high in sugar and refined carbohydrates can lead to inflammation [38], which may increase the risk of certain types of cancer [39]. Further studies are needed to determine the precise functions of the intestinal microbiota in causing intestinal dysplasia and/or cancer development, especially for the pathways mentioned above that have not been previously reported.

This study demonstrated that ALGD patients have a characteristic gut microbial composition. However, there were some limitations to this study. First, the mucosal microbiota in the intestinal epithelium and the luminal microbiota in the stool are markedly different ecosystems, which may have had different effects on the initiation and development of ALGD/CRC. Therefore, the comparison between the microbiota in both ecosystems is required in future studies. Second, owing to the limitations of the current microbiological detection methods and microbiota database (particularly the fungal database), we could not annotate some of the bacteria and fungi at the species level. Third, ALGD is the precancerous condition of CRC. Thus, the microecological changes in the ALGD group are not as apparent as those seen in the CRC group. Therefore, a study with a larger sample size and a more comprehensively designed experimental protocol, which can observe the microecological changes in the entire process of polyp-adenoma-colorectal carcinoma, is required.

Intestinal mucosal microbiota, including bacteria and fungi, has been shown to play a critical role in the initiation and development of colorectal cancer. In addition, recent studies suggest that the mucosal microbiota may be a more important characteristic of early-onset colorectal cancer than luminal microbiomes [40]. These findings indicate that the mucosal microbiota metabolic pathways related could serve as a potential marker for the diagnosis and treatment of colorectal adenoma/carcinoma.

Correlation and functional prediction analyses have provided insights into the complex direct and/or indirect pathways through which the intestinal microbiome affects host gut inflammation and/or metabolic pathways [41]; Future studies should focus on identifying specific microorganisms or metabolites with carcinogenic or cancer-suppressing effects, as well as the inter- and intra-kingdom interactions between intestinal bacteria and fungi that contribute to the development of intestinal dysplasia and/or cancer. Furthermore, we need to expand our knowledge of the effects of the intestinal microbiota on host immune and metabolic systems, which could significantly influence the development of colorectal adenoma/carcinoma. By comprehending the mechanisms that underlie the relationship between the microbiota and host factors, we can devise new strategies for preventing and treating colorectal cancer.

## Figures and Tables

**Figure 1 microorganisms-11-01327-f001:**
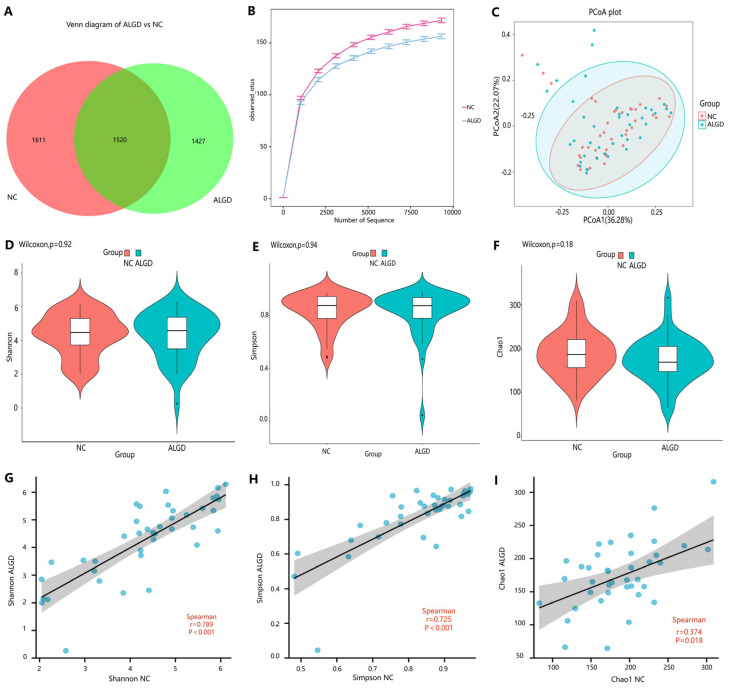
Colonic mucosal bacterial richness and diversity in adenomas with low-grade dysplasia (ALGD) and normal control (NC) tissues. Bacterial composition in the ALGD tissues was compared with the adjacent NC. (**A**) Bacterial operational taxonomic units (OTUs) number in ALGD mucosa and NC in the colon. (**B**) Rarefaction analysis of sampling by observed bacterial OTU. (**C**) Principal coordinates analysis (PCoA) based on the relative abundance of bacterial OTUs in ALGD and NC. (**D**) Shannon, (**E**) Simpson, and (**F**) Chao 1 index between ALGD mucosa and NC in the colon. The ecologic associations of (**G**) Shannon, (**H**) Simpson, and (**I**) Chao 1 index between ALGD and NC groups. Each dot represent one subject; gray area represents 95% confidence interval for best-fit line.

**Figure 2 microorganisms-11-01327-f002:**
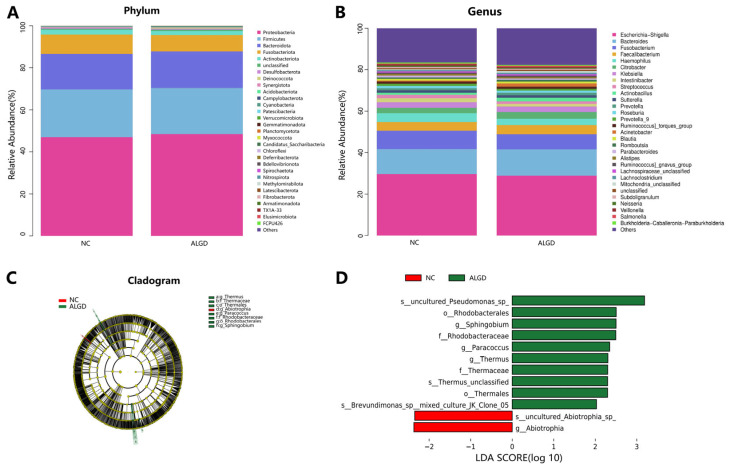
Characterization of microbiome distribution in patients with adenomas with low-grade dysplasia (ALGD). Relative abundance of bacterial phyla (**A**) and genera (**B**) in the ALGD and NC groups. (**C**) Taxonomic representation of statistically and biologically consistent differences between colonic ALGD and NC groups. (**D**) Histogram of the LDA scores (log10) computed for features differentially abundant in colonic ALGD and NC.

**Figure 3 microorganisms-11-01327-f003:**
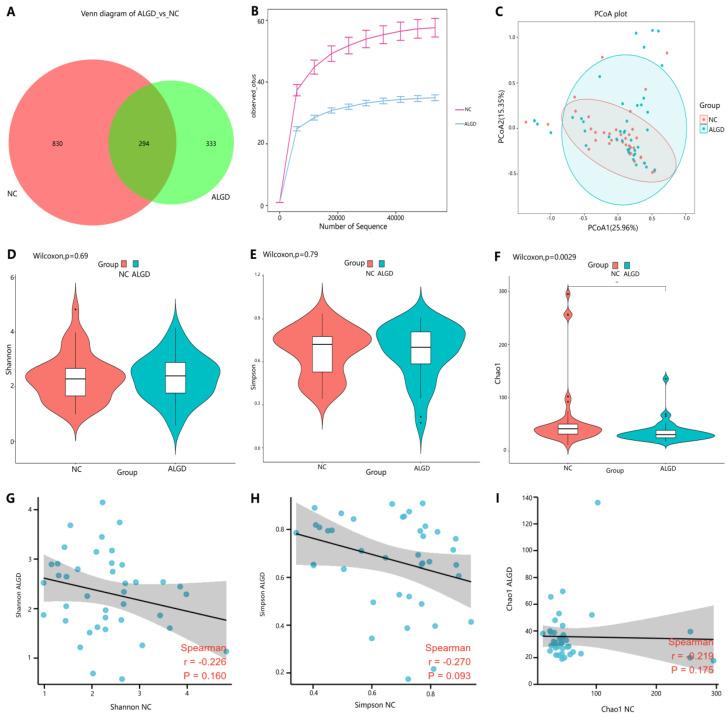
Colonic mucosal fungal richness and diversity in adenomas with low-grade dysplasia (ALGD) and normal control (NC) tissues. Fungal composition in the ALGD tissues was compared with the adjacent NC. (**A**) Fungal operational taxonomic units (OTUs) number in ALGD mucosa and NC in the colon. (**B**) Rarefaction analysis of sampling by observed Fungal OTU. (**C**) Principal coordinates analysis (PCoA) based on the relative abundance of Fungal OTUs in ALGD and NC. (**D**) Shannon, (**E**) Simpson, and (**F**) Chao 1 index between ALGD mucosa and NC in the colon. The ecologic associations of (**G**) Shannon, (**H**) Simpson, and (**I**) Chao 1 index between ALGD and NC groups.Each dot represent one subject; gray area represents 95% confidence interval for best-fit line.

**Figure 4 microorganisms-11-01327-f004:**
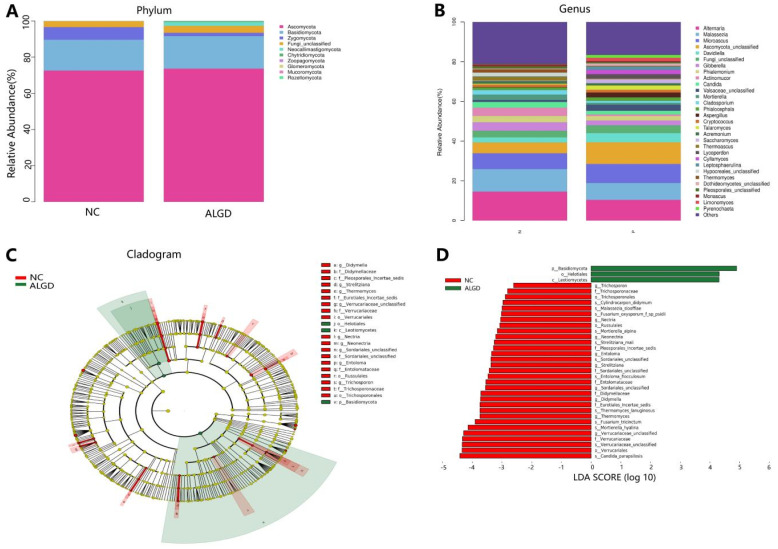
Characterization of fungal distribution in patients with adenomas with low-grade dysplasia (ALGD). Relative abundance of fungal phyla (**A**) and genera (**B**) in the ALGD and NC groups. (**C**) Taxonomic representation of statistically and biologically consistent differences between the colonic ALGD and NC groups. (**D**) Histogram of the LDA scores (log10) computed for features differentially abundant in colonic ALGD and NC groups.

**Figure 5 microorganisms-11-01327-f005:**
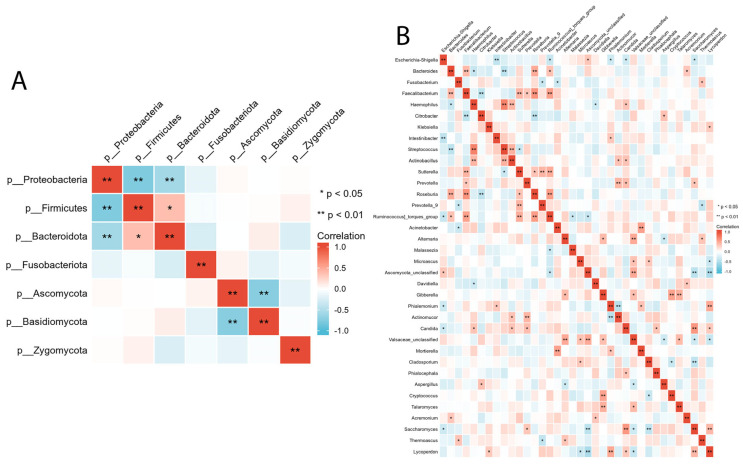
Inter- and intra-kingdom correlations between intestinal bacteria and fungi at phylum (**A**) and genus (**B**) level. (* *p* < 0.05, ** *p* < 0.01).

**Figure 6 microorganisms-11-01327-f006:**
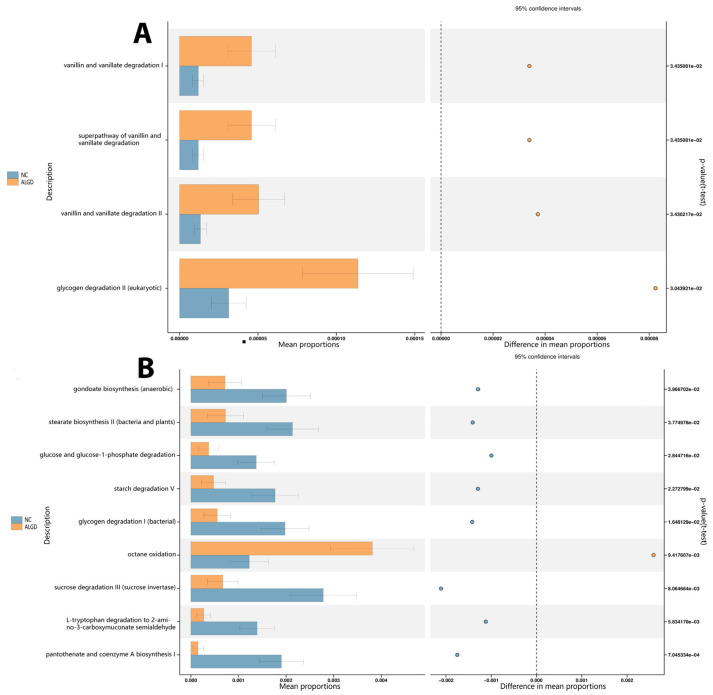
Significant metagenomic differences were found in bacterial (**A**) and fungal (**B**) functional analysis by Phylogenetic Investigation of Communities by Reconstruction of Unobserved States (PICRUSt2), respectively, between the ALGN and NC groups. In addition, a significant difference was detected by stamp v2.1.3 using White’s non-parametric *t* test (*p* < 0.05). The left figure in the picture shows the abundance ratio of different functions in two groups of samples. The dots represent the proportion of difference in functional abundance within the 95% confidence interval. The right value is the *p*-value.

## Data Availability

Data from this study are available from the corresponding author upon reasonable request.

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
