# Peer review of "Alteration of Colonic Bacterial and Fungal Composition and Their Inter- and Intra-Kingdom Interaction in Patients with Adenomas with Low-Grade Dysplasia"

_microorganisms, 2023, doi:10.3390/microorganisms11051327_

Round 1

Reviewer 1 Report

The introduction and discussion required more information and analysis of the results as well as bibliographic support.

Material and methods:

1)     Explain if the biopsy tissues were immediately stored at -80°C or placed in a solution to preserve the specimen and the genetic material.

2)     Inclusion and exclusion criteria should go before indicating the number of subjects and biopsies.

3)      What was the concentration of the primers used?

4)      Indicate the quantity of DNA with which the Illumina NovaSeq platform analysis was performed.

5)      Define operational taxonomic units (OTUs) in the body of the manuscript.

6)     In the legend of Figure 1, it was not indicated that it is a diversity analysis.

7)      Explain why in Figure 1G-I, there is a negative correlation and in discussion became positive.

8)     Correct the wording for “newoperational taxonomic unit (OTUs)”.

9)     Explain why there is no correlation in Figures (1, 3), of "(97% similarity level)" which is not reflected in both figures.

10)   There are no fungal bacteria, and this paragraph talks about fungi, not bacteria page 7 as said “However, no significant differences were found in the major bacteria phyla and genera (Figure 4B).”

11)   The definition of the images is too poor, Figure 4C, requires improvement.

12)   The legend of figure 4 says "Relative abundance of bacterial phyla (A)", and it should say fungi.

13)   In the discussion, the authors describe poorly their results and there is not a deep analysis of them.

Author Response

Dear Editor,

Thank you very much for your decision letter and advice on our manuscript.We also thank the reviewers for the constructive and thoughtful comments and suggestions. Accordingly, we have revised the manuscript. All amendments are highlighted in red in the revised manuscript. In addition, point-by-point responses to the comments are listed below this letter.

We hope that the editors and reviewers will be satisfied with our modifications and the paper is now suitable for publication in your journal.

Thank you for consideration our manuscript revision. We are looking forward to hearing from you soon!

Xinyun Qiu

Material and methods:

  • Explain if the biopsy tissues were immediately stored at -80°C or placed in a solution to preserve the specimen and the genetic material.

Response: thank you for your question. The specimens from polyps and the adjacent normal control mucosa were biopsied and subsequently stored at -80 °C for further analysis, we have added these content in the method section in detail.

  • Inclusion and exclusion criteria should go before indicating the number of subjects and biopsies.

Response: thank you very much for your good suggestion. We have revised the manuscript according to your suggestion.

  • What was the concentration of the primers used?

Response:Thank you very much for your kind question. We used a primer concentration of 1 μM, and we have added this information to the Method section. To make it easier to locate, we have highlighted the relevant content in red.

  • Indicate the quantity of DNA with which the Illumina NovaSeq platform analysis was performed.

Response: Thank you very much for your kind question. We used the Illumina NovaSeq 6000 high-throughput platform for the 16S rRNA and ITS1-2 gene sequence detect.We have highlighted the relevant content in the method section in red.

  • Define operational taxonomic units (OTUs) in the body of the manuscript.

Response:Thank you very much for your kind question. The operational taxonomic units (OTUs) were defined using a 97% similarity cut-off value.We have added and highlighted the relevant content in the method section in red.

  • In the legend of Figure 1, it was not indicated that it is a diversity analysis.

Response:Thank you very much for your kind question. We have indicated that it is a diversity analysis in the figure legend 1 and highlighted the relevant content in the method section in red.

  •      Explain why in Figure 1G-I, there is a negative correlation and in discussion became positive.

Response: Thank you for pointing out our mistake, we have corrected this mistake.There is positive correlations in Figure 1G-I.

  • Correct the wording for “newoperational taxonomic unit (OTUs)”.

Response:  Thank you very much for pointing out our mistake. Actually, there should be a space between “new” and “operational”, and we have corrected this mistake.

  • Explain why there is no correlation in Figures (1, 3), of "(97% similarity level)" which is not reflected in both figures.

Response: thank you very much for your kind question. We would like to clarify that the 97% similarity level mentioned in the figure legend refers to the OTUs that were defined using a 97% similarity cut-off value, which we have previously described in the method section. To avoid any confusion, we have removed the "(97% similarity level)" from the figure legend.

  • There are no fungal bacteria, and this paragraph talks about fungi, not bacteria page 7 as said “However, no significant differences were found in the major bacteria phyla and genera (Figure 4B).”
  • Response: Thank you very much for pointing out our mistake. The sentence should be “However, no significant differences were found in the major fungal phyla and genera (Figure 4B).”we have corrected this mistake.

  •  The definition of the images is too poor, Figure 4C, requires improvement.

Response: Thank you for bringing this deficiency to our attention. According to your kind advice, we have improved the definition of this image.

  • The legend of figure 4 says "Relative abundance of bacterial phyla (A)", and it should say fungi.

Response: Thank you very much for pointing out our mistake. We have corrected this mistake.

  • In the discussion, the authors describe poorly their results and there is not a deep analysis of them.

Response: Thank you very much for your kind suggestion. We have substantially revised the Discussion section to bring the discussion to greater depth and highlighted the relevant content in the discussion section in red.

Reviewer 2 Report

The manuscript titled: “Alteration of colonic bacterial and fungal composition and their inter- and intra-kingdom interaction in patients with adenomas with low-grade dysplasia” shows novel information about microbiota characterization from patients presenting low-grade dysplasia. Authors should add line numbers to ease the reviewing. The manuscript presented high similarity with previously published articles, so authors are encouraged to reduce the similarity level:

Lin et al. (2022). https://doi.org/10.1155/2022/3140070

Kuo et al. (2022). https://doi.org/10.3390/microorganisms11020234

Qin et al. (2022). https://doi.org/10.3389/fnut.2022.819980

Saito et al. (2019). https://doi.org/10.1371/journal.pone.0212406

Abstract

1.     What is the importance of having gut microbiota signatures of patients with colorectal adenomas? This should be stated in the abstract.

2.     The abstract should go beyond the enumeration of species. What is the meaning of having those species in a higher or lower concentration?

3.     The conclusion seems obvious (fungal and microbial composition is altered).

Background

4.     A citation is needed for the sentence: “(…) However, the luminal (…) immune system (…)”.

5.     Please delete “etc.” from: “(…) drug usage, etc.” as this is not a proper scientific expression. Authors should use a different one.

6.     The authors indicated that microbiota findings are different due to several factors affecting it. How does this study guarantee that results are consistent, considering these factors? This should be stated by the authors.

Methods

7.     Please avoid starting sentences with numbers in: “55 subjects (aged between …)”

8.     Please make a diagram for the Patients and Biopsy sample collection. This would significantly facilitate the understanding of this section.

9.     Demographic characteristics of the patients are needed.

10.  Other than the gel concentration, what conditions were used to assess DNA quality through electrophoresis? (e.g., voltage and running time).

11.  Please write the gene names using italics: 16SrRNA

Results

12.  How do the authors explain that no differences in the microbial profiles were found compared to other research groups?

13.  The authors have indicated that fungi have been overlooked in previous studies. What do the authors hypothesize why has this happened?

14.  A citation is needed in the sentence: “(…) diets with high sugar and refined carbohydrates can lead to inflammation”. Is there a direct link between high sugar diets-inflammation-colon cancer, or are the authors just indicating that a higher inflammation could be associated with a higher cancer risk?

15.  The authors did not find most changes in early cancer stages. However, there is evidence suggesting this situation. Why did the authors choose early cancer stages instead of advanced cancer?

16.  What are the perspectives of this work? The authors vaguely indicated the potential applications at the end of the discussion, but more specific and cited applications are needed.

Author Response

Dear Editor,

Thank you very much for your decision letter and advice on our manuscript.We also thank the reviewers for the constructive and thoughtful comments and suggestions. Accordingly, we have revised the manuscript. All amendments are highlighted in red in the revised manuscript. In addition, point-by-point responses to the comments are listed below this letter.

We hope that the editors and reviewers will be satisfied with our modifications and the paper is now suitable for publication in your journal.

Thank you for consideration our manuscript revision. We are looking forward to hearing from you soon!

Xinyun Qiu

Abstract

  1.   What is the importance of having gut microbiota signatures of patients with colorectal adenomas? This should be stated in the abstract.

Response: Thank you very much for you good suggestion. We have modified the abstract and we have highlighted the relevant content in the abstract section in red.

  1.  The abstract should go beyond the enumeration of species. What is the meaning of having those species in a higher or lower concentration?

Response:Thank you very much for you good suggestion. We have modified the abstract and we have highlighted the relevant content in the abstract section in red. Due to the space limitations, we do not describe the characteristics of the various different microbiota in the abstract. However, we discussed them in detail in our discussion section.

  1. The conclusion seems obvious (fungal and microbial composition is altered).

Response: Thank you very much for you kind suggestion. We have added the significance of this study in the abstract.

Background 

  1. A citation is needed for the sentence: “(…) However, the luminal (…) immune system (…)”.

Response:Thank you very much for your kind suggestion. We have added the citation to this sentence. 

  1. Please delete “etc.” from: “(…) drug usage, etc.” as this is not a proper scientific expression. Authors should use a different one.

Response:Thank you very much for you kind suggestion. We have changed the wording of this sentence.

  1. The authors indicated that microbiota findings are different due to several factors affecting it. How does this study guarantee that results are consistent, considering these factors? This should be stated by the authors.

   Response:This is an excellent question.Although controlling all factors that can affect gut microbiota is challenging, we established strict inclusion and exclusion criteria when enrolling subjects, as described by Qiu et al. (Therapeutic Advances in Gastroenterology, 2020) in the Methods section. In addition, this study employed a self-control design, where dysplasia and normal control samples were collected from the same subject. This design helped to eliminate more confounding factors, such as age, sex, and diet, resulting in more consistent results.

Methods 

  1. Please avoid starting sentences with numbers in: “55 subjects (aged between …)”

Response:Thank you very much for your kind suggestion. We have corrected the mistake.

  1. Please make a diagram for the Patients and Biopsy sample collection. This would significantly facilitate the understanding of this section.

Response: Thank you very much for your kind suggestion. We have added the diagram in the manuscript.

  1. Demographic characteristics of the patients are needed.

Response:Thank you very much for your kind suggestion. We have described characteristics of the patients in the result section.

  1. Other than the gel concentration, what conditions were used to assess DNA quality through electrophoresis? (e.g., voltage and running time).

Response:Thank you very much for your kind suggestion.The PCR products were subsequently verified through 2% agarose gel electrophoresis ( run at a constant voltage of 120 V for 40 min). we have added the content in the method section and marked the content in red.

  1. Please write the gene names using italics: 16SrRNA

Response: Thank you very much for your kind suggestion. We have followed your advice and modified the manuscript followed by your advice.

Results 

  1. How do the authors explain that no differences in the microbial profiles were found compared to other research groups?

Response: Thank you very much for your good question. The differences in microbiota profiles observed in our study were not as significant as those reported in previous studies between colorectal cancer (CRC) and non-cancer (NC) groups (Science 2019, 364, 1133-1135). This discrepancy could be partly attributed to our study's self-control design, whereas most of the other studies employed non-self-control designs. However, there were still some differences between the ALGD and NC groups. For example, ten taxa, including uncultured Pseudomonas spp., Rhodobacterales, Sphingobium, Rhodobacteraceae, Paracoccus, Thermus, Thermaceae, and Thermus unclassified species, and Thermales order species were increased in ALGD group, while two taxa (an uncultured_Abiotrophia_species and Abiotrophia genus) were decreased in ALGD group compared with NC. Among the twelve taxa, several taxa colonize the human gut. Wheatley et al.[Nat Commun 2022,13, 6523] reported that a Pseudomonas species (P. aeruginosa) can translocate from the gut to the lungs and cause tissue damage during lung infection with greater pro-inflammatory gene expression. Mei et al. [Exp Ther Med 2018, 16, 856-866] reported that Sphingobium was more abundant in the duodenal mucosa of patients with pancreatic cancer than in the duodenal microbiotas of healthy controls. As for Abiotrophia, one study reported its lower abundance in colonic mucosal tissue in high-fat-diet subjects[Exp Ther Med 2018, 16, 856-866,], and another study reported that it is more abundant in the tongue coating microbiome in patients with gastric cancer[J Cancer 2018, 9, 4039-4048,]. Previous studies have demonstrated the pro-inflammatory function of these bacterial taxa in human disorders; however, the relationship between these bacterial taxa and ALGD development need further study. We have discussed these content in the discussion section.

  1. The authors have indicated that fungi have been overlooked in previous studies. What do the authors hypothesize why has this happened?

  1. Response: Thank you very much for your good question. We hypothesize that fungi have been overlooked in previous studies for several reasons, including the fact that fungi are more difficult to identify and characterize than bacteria, and that most studies have focused on bacterial communities in the gut due to their abundance and established roles in gut health and disease. As a result, fungi may have been underestimated or ignored in previous studies, despite their potential importance in gut health and disease.We have add the content inthe background section.

  1. A citation is needed in the sentence: “(…) diets with high sugar and refined carbohydrates can lead to inflammation”. Is there a direct link between high sugar diets-inflammation-colon cancer, or are the authors just indicating that a higher inflammation could be associated with a higher cancer risk?

Response: Thank you very much for your reminding. We have added the citation which reported that the high sugar diet can contribute to both intestinal inflammation and colon cancer.

  1. The authors did not find most changes in early cancer stages. However, there is evidence suggesting this situation. Why did the authors choose early cancer stages instead of advanced cancer? 

Response: Thank you very much for your good question. As there have been many studies on the gut microbiota in advanced colorectal cancer, but few studies have compared the microbiota between colonic mucosal low-grade dysplasia and normal mucosa, especially with a self-control design. We aimed to investigate this aspect. Furthermore, many colorectal cancers originate from colonic low-grade dysplasia. Interestingly, we have identified several specific bacteria and fungi taxa and metabolic pathways are changed in the ALGD mucosal samples and the normal control, these changes may serve as potential markers for the diagnosis and treatment of colorectal adenoma and carcinoma.

  1. What are the perspectives of this work? The authors vaguely indicated the potential applications at the end of the discussion, but more specific and cited applications are needed. 

Response: Thank you very much for your kind suggestion. In this study, we identified the different composition of intestinal bacterial and fungal composition between ALGD and NC. And hypothesized several metabolite could participate in the development of colonic adenoma and carcinoma. Future studies should focus on identifying specific microorganisms or metabolites with carcinogenic or cancer-suppressing effects, as well as the inter- and intra-kingdom interactions between intestinal bacteria and fungi that contribute to the development of intestinal dysplasia and/or cancer. Furthermore, there is a need to expand our knowledge of the effects of the intestinal microbiota on host immune and metabolic systems, which could significantly influence the development of colorectal adenoma/carcinoma. By comprehending the mechanisms that underlie the relationship between the microbiota and host factors, we can devise new strategies for the prevention and treatment of colorectal cancer. We have improved the discussion section thoroughly and marked the modified content in red.

Reviewer 3 Report

This study is designed to characterize the microbiota composition of ALGD and normal colorectal mucosa tissues. Authors have used 16S and ITS1-2 rRNA gene sequencing and bioinformatics tools to analyze a total of 40 patients to explore inter and intra-kingdom correlations between intestinal bacteria and fungi. The results of this study detected altered bacterial and fungal composition in the ALGD group compared to normal mucosa. Several metabolic pathways were also found to be modified during bacterial and fungal functional analysis.

It's a well-done systematic study however the analysis has been performed in ALGD samples and not CRC it does not provide a direct correlation between mucosal microbiota changes and the development of CRC. It would be great if authors could use CRC samples from a larger population to establish a close link between gut microbiota and CRC pathogenesis.

Author Response

This study is designed to characterize the microbiota composition of ALGD and normal colorectal mucosa tissues. Authors have used 16S and ITS1-2 rRNA gene sequencing and bioinformatics tools to analyze a total of 40 patients to explore inter and intra-kingdom correlations between intestinal bacteria and fungi. The results of this study detected altered bacterial and fungal composition in the ALGD group compared to normal mucosa. Several metabolic pathways were also found to be modified during bacterial and fungal functional analysis.

It's a well-done systematic study however the analysis has been performed in ALGD samples and not CRC it does not provide a direct correlation between mucosal microbiota changes and the development of CRC. It would be great if authors could use CRC samples from a larger population to establish a close link between gut microbiota and CRC pathogenesis.

Response: Thank you so much for your recognition and positive evaluation of our article. As there have been many studies on the gut microbiota in advanced colorectal cancer, but few studies have compared the microbiota between colonic mucosal low-grade dysplasia and normal mucosa, especially with a self-control design. We aimed to investigate this aspect. Furthermore, many colorectal cancers originate from colonic low-grade dysplasia. Interestingly, we have identified several specific bacteria and fungi taxa and several metabolic pathways that could be changed in the ALGD mucosal samples and the normal control, these changes may serve as potential markers for the diagnosis and treatment of colorectal adenoma and carcinoma.

Reviewer 4 Report

Manuscript:  Alteration of colonic bacterial and fungal composition and their inter- and intra-kingdom interaction in patients with adenomas with low-grade dysplasia

General Comments

This manuscript examined the microbial composition colorectal adenoma and normal mucosal tissue collected from 40 patients diagnosed with adenoma with low-grade dysplasia.  This is an interesting study, a few specific comments are included below.

Specific Comments

1.  Results, Global shifts in the colonic microbes, page 4:  In the first sentence, the authors state that metagenomic shotgun sequencing was used but the Methods section describes 16S sequencing.  Please clarify.

2.  Results, Global shifts in the colonic microbes, page 4:  In the text, the authors state that the “…Chao1, Simpson, and Shannon index in the ALGD group was negatively correlated with that in the NC” but Figures 1G-I do not seem to reflect this.  Could the authors please explain?

3.  Results:  Perhaps there are limitations in sample size, but did the authors consider further investigating microbial composition by polyp location?

4.  Discussion:  It would be helpful to also include the limitations of 16S sequencing methods in the limitations section.

Author Response

General Comments

This manuscript examined the microbial composition colorectal adenoma and normal mucosal tissue collected from 40 patients diagnosed with adenoma with low-grade dysplasia.  This is an interesting study, a few specific comments are included below.

Response: Thank you so much for your recognition and positive evaluation of our article, which will encourage us to do more research on the gut microbiota in the development of colorectal adenoma and turmor.

Specific Comments

  1. Results, Global shifts in the colonic microbes, page 4:  In the first sentence, the authors state that metagenomic shotgun sequencing was used but the Methods section describes 16S sequencing.  Please clarify.

Response: Thank you very much for pointing out this mistake, actually, it is 16s sequencing. We have corrected this mistake.

  1. Results, Global shifts in the colonic microbes, page 4:  In the text, the authors state that the “…Chao1, Simpson, and Shannon index in the ALGD group was negatively correlated with that in the NC” but Figures 1G-I do not seem to reflect this.  Could the authors please explain?

Response: Thank you very much for pointing out this mistake, it should be “positively correlated”, and we have corrected this mistake.

  1. Results:  Perhaps there are limitations in sample size, but did the authors consider further investigating microbial composition by polyp location?

Response:  Thanks a lot for your good suggestion. Due to the limitation in sample size, we did not further investigating the microbial composition by polyp location. And we will do further research on this area in our future investigation based on a larger sample size.

  1. Discussion:  It would be helpful to also include the limitations of 16S sequencing methods in the limitations section.

Response: Thank you very much for your good suggestion. We have added the limitations of microbiological detection methods in the limitations section.

Reviewer 5 Report

This manuscript "Alteration of colonic bacterial and fungal composition and their inter- and intra-kingdom interaction in patients with adenomas with low-grade dysplasia" although it has some limitations described by the same authors in the discussion, in my opinion it is useful to the scientific community because it puts relationship the microbiota with : Colorectal cancer and colorectal adenomas with low-grade dysplasia, two different stages of colon cancer so the microbiota can be a tumor marker or a potential drug target. This contributes to expanding the weapons available to combat this pathology. The manuscript is complete in all its sections.

Author Response

This manuscript "Alteration of colonic bacterial and fungal composition and their inter- and intra-kingdom interaction in patients with adenomas with low-grade dysplasia" although it has some limitations described by the same authors in the discussion, in my opinion it is useful to the scientific community because it puts relationship the microbiota with : Colorectal cancer and colorectal adenomas with low-grade dysplasia, two different stages of colon cancer so the microbiota can be a tumor marker or a potential drug target. This contributes to expanding the weapons available to combat this pathology. The manuscript is complete in all its sections.

Response: Thank you so much for your recognition and positive evaluation of our article, which will encourage us to do more research on the gut microbiota in the development of colorectal adenoma and cancer.

Reviewer 6 Report

This manuscript was very interesting. Moreover, i have some suggestions:

Abstract solido be shorter;
Introduction was short!
Statistical analisi su can include correlation analysis;
References should be increased.

Author Response

This manuscript was very interesting. Moreover, i have some suggestions:

Abstract solido be shorter;

Introduction was short!

Statistical analisi su can include correlation analysis;

References should be increased.

Response: Thank you so much for your recognition and positive evaluation of our article. We have shortened the abstract, supplemented the background section of the article, include correlation analysis in the statistical analysis, and add some references followed by your advice. Thanks again for your kind help.

Round 2

Reviewer 1 Report

The manuscript has been improved substantially and now can be accepted for publication.